# Alterations of the Composition and Neurometabolic Profile of Human Gut Microbiota in Major Depressive Disorder

**DOI:** 10.3390/biomedicines10092162

**Published:** 2022-09-02

**Authors:** Alexey S. Kovtun, Olga V. Averina, Irina Y. Angelova, Roman A. Yunes, Yana A. Zorkina, Anna Y. Morozova, Alexey V. Pavlichenko, Timur S. Syunyakov, Olga A. Karpenko, George P. Kostyuk, Valery N. Danilenko

**Affiliations:** 1Vavilov Institute of General Genetics, Russian Academy of Sciences (RAS), 119333 Moscow, Russia; 2Skolkovo Institute of Science and Technology, 121205 Moscow, Russia; 3Moscow Institute of Physics and Technology, State University, 141701 Dolgoprudny, Russia; 4Mental-Health Clinic No. 1 Named after N.A. Alexeev of Moscow Healthcare Department, 117152 Moscow, Russia; 5Department Basic and Applied Neurobiology, V.P. Serbsky Federal Medical Research Centre of Psychiatry and Narcology, 119034 Moscow, Russia; 6Federal State Budgetary Educational Institution of Higher Education, Moscow State University of Food Production, 125080 Moscow, Russia; 7Caspian International School of Medicine, Caspian University, 521 Seifullin Street, Almaty 050000, Kazakhstan

**Keywords:** depression, gut microbiota, gut–brain axis, taxonomic composition, whole metagenome, orthologs, neuroactive metabolites, signature, dysbiosis, biomarkers

## Abstract

Major depressive disorder (MDD) is among the most prevalent mental disorders worldwide. Factors causing the pathogenesis of MDD include gut microbiota (GM), which interacts with the host through the gut–brain axis. In previous studies of GM in MDD patients, 16S rRNA sequencing was used, which provided information about composition but not about function. In our study, we analyzed whole metagenome sequencing data to assess changes in both the composition and functional profile of GM. We looked at the GM of 36 MDD patients, compared with that of 38 healthy volunteers. Comparative taxonomic analysis showed decreased abundances of *Faecalibacterium prausnitzii*, *Roseburia hominis*, and *Roseburia intestinalis*, and elevated abundances of *Escherichia coli* and *Ruthenibacterium lactatiformans* in the GM of MDD patients. We observed decreased levels of bacterial genes encoding key enzymes involved in the production of arginine, asparagine, glutamate, glutamine, melatonin, acetic, butyric and conjugated linoleic acids, and spermidine in MDD patients. These genes produced signature pairs with *Faecalibacterium prausntizii* and correlated with decreased levels of this species in the GM of MDD patients. These results show the potential impact of the identified biomarker bacteria and their metabolites on the pathogenesis of MDD, and should be confirmed in future metabolomic studies.

## 1. Introduction

According to the World Health Organization, depression is one of the most prevalent mental health disorders, the second leading cause of disability worldwide, and a major contributor to deaths by suicide. Approximately 4.4% of the world’s population is affected by depression [1]. The rise in global COVID-19 rates has been accompanied by an increase in the prevalence of significant neuropsychiatric disorders such as depression, which can result from social stressors such as isolation and unemployment [2,3]. This public health problem is considered a global psychological pandemic [4], and has encouraged investment into research on mental wellbeing [5]. Major depressive disorder (MDD) is a common psychiatric illness, and typical symptoms include depressed mood and/or loss of interest or pleasure in life activities for at least two weeks. Symptoms also include unintentional weight change, insomnia or hypersomnia, agitation or psychomotor retardation, fatigue, feelings of worthlessness or guilt, and attempting suicide. Thus, MDD significantly reduces quality of life and has become a serious medical and social problem.

The mechanisms of disease development remain unclarified, as it has a heterogeneous etiology. It has been shown that genetics, neuro-endocrinology, neuro-immunity, and structural and functional disorders of the brain all contribute to the pathophysiology of MDD. Mechanisms that have been found to be linked to depression also include a dysfunctional hypothalamic-pituitary-adrenal (HPA) axis [6], immune-inflammatory and oxidative pathways [7], altered vagus nerve tone [8], and imbalance between neural excitatory and inhibitory signaling [9]. Other factors that can lead to the development of depression includes chronic and acute stress [10], depletion of monoamines [11], poor nutrition [12], mitochondrial dysfunction [13], disrupted metabolism [14], environmental factors and epigenetics [15], infection [16], levels of sex hormones [17], etc. Inflammation is now considered one of the primary pathologies that lead to depression. Various theories of inflammation, such as the macrophage hypothesis or cytokine theory, have been proposed as the main pathology in MDD. The main process when suppressing inflammation is the activation of the immune response, in particular, the production of cytokines. This affects the levels of neuro-compounds, which in turn leads to the development of MDD. Recently, a novel kynurenine pathway (KP) has drawn attention in the cytokine theory [10]. Pro-inflammatory cytokines activate the KP by affecting tryptophan (TRP) metabolism and releasing neurotoxins, which can either reduce serotonin production or promote serotonin reuptake [13]. The COVID-19 pandemic has led to long-term psychiatric symptoms, possibly due to an immunological reaction to the virus itself, as well as fundamental changes in life associated with the pandemic. The immune response can lead to changes in the functioning of the tryptophan–kynurenine pathway, which plays an important role in the pathophysiology of mental illness [18]. The monoamine hypothesis has also been a prevailing theory for the pathogenesis of depression. This hypothesis holds that depression is caused by the depletion of 5-HT, norepinephrine, or dopamine in the central nervous system [19].

Depression is associated with an increased risk of atherosclerosis, cardiovascular diseases, hypertension, stroke, dementia, neurodegenerative diseases, as well as metabolic disorders, such as type 2 diabetes [20]. Depressive symptoms are also observed during physical conditions, such as cancer, autoimmune diseases, or systemic infections, in which chronic inflammation has been implicated [21]. 

Today, gut microbiota (GM)—which interacts with the host through the gut–brain axis (GBA)—is considered an important factor associated with the pathology of depression. Clinical and experimental data indicate a crosstalk between microbiota and intestinal cells and the nervous system, as well as between microbiota and the brain through metabolic, neuroendocrine, and neuroimmune pathways. Increasing evidence suggests a close relationship between MDD and the dysfunction of the GBA. Evidence from both animal and clinical studies support the pivotal role of GM on mood modulation. For example, germ-free mice exhibited decreased anxiety and depression-like behavior. However, following the transplantation of flora from MDD patients, the mice began exhibiting the depressive phenotype [22].

GM is a major source of metabolites, and can affect the host in various ways, such as through vagus nerve stimulation, changes in central neurotransmission, modulation of systemic and neuroinflammation, and crossing the blood–brain barrier and binding to receptors in the brain [23]. Some studies indicate an abnormal production of GM metabolites, manifesting from various neuropsychiatric disorders [24,25]. GM exerts its effects through immune system activation (e.g., inflammatory cytokines and chemokines) and production of neurotransmitters and other neurometabolites (e.g., serotonin, gamma-aminobutyric acid (GABA) and glutamate (GLU), short-chain fatty acids (SCFAs), and key amino acids, such as TRP) [26]. Neurotransmitters synthesized by bacteria stimulate the secretion of molecules by specific intestinal epithelial cells, for example, enterochromaffin cells, which are responsible for signal transmission through the enteric nervous system [21,27]. Bacterial products participate in stimulation of central receptors, peripheral stimulation of neural, endocrine, and immune mediators, and epigenetic regulation of histone acetylation and DNA methylation, which have been implicated in depression [26]. 

Some studies suggested that altered gut microbe levels in conditions causing dysbiosis are associated with changed brain structure [28]. Dysbiosis is accompanied by a decrease in the number of *Faecalibacterium prausnitzii* and *Roseburia intestinalis* bacteria, known to produce butyrate [29]. In addition, increased abundances of opportunistic pathogens, such as *Bacteroides caccae*, *Clostridium hathewayi*, *Clostridium symbiosum*, *Erysipelatoclostridium ramosum*, and *Escherichia coli*, has also been related to dysbiosis [30]. Such changes can be a potential cause of inflammation in the brain through blood circulation and the increased production of various cytokines, including IL-6, IL-1β, and TNF-α [31], which also modulate processes in the brain that affect mood and behavior [32].

Human GM is a complex community of microorganisms largely composed of anaerobic and facultative anaerobic bacteria that significantly impact human health and wellbeing [33]. It consists of 10^14^ to 10^15^ bacterial cells, and, to date, more than 1000 species have been identified as part of GM [34,35]. The largest number of commensal bacteria is detected in the colon. GM participates in the fermentation of food polysaccharides and proteins, synthesis of vitamins, and enzymatic transformation of endogenous metabolites, such as bile acids [36]. GM also produces microbial compounds that mediate crosstalk with the immune system of the host and play an important role in the GBA [37]. Published data on the composition of the GM of depressive patients are frequently contradictory, and these discrepancies could be explained by the broad spectrum of MDD manifestations, geographical differences between participants (which may result in genetic and/or dietary differences), potential sub-types of GM within MDD groups, inadequate statistics for testing, and different molecular technologies used to investigate stool samples [26].

To determine the microbial community composition of patients with depression, previous studies have mainly used 16S rRNA gene sequencing [38]. However, a limitation of this method is that it only provides information on the taxonomic composition of the studied specimens and cannot directly assess the functions of the microbiota. In this work, we used whole metagenome sequencing analysis, which gave us an opportunity to profile functional representations and metabolic pathways in the bacterial community. Application of the developed algorithm for metagenomic analysis allowed us to link GM neurometabolic capacity with depression.

## 2. Materials and Methods

### 2.1. Selection of the Cohorts of Patients with Depression and Healthy Volunteers

The cohorts used for the research included volunteers aged 18 to 54 years old from Moscow or the Moscow region.

Patients with depressive outbreaks (group PwD) of medium or high severity (codes F31.3, F31.4, F31.5, F32.1, F32.2, F32.3, F32.8, F32.9, F33.1, F33.2, and F33.3, according to International Classification of Diseases Version 10, ICD-10) were selected among those admitted for inpatient treatment at Psychiatric Hospital No. 1 named after N. A. Alekseyev of the Department of Health in Moscow. Initial examination and history documentation were performed in accordance with institutional clinical practice, including a physical examination (height, weight, and body mass index (BMI)). Patient psychological evaluation was conducted in two stages. First, patients filled out two questionnaires that allowed estimation of their condition: the Center for Epidemiological Studies Depression scale (CES-D) and the 7-item Generalized Anxiety Disorder questionnaire (GAD-7). Next, a psychiatrist administered the 17-item Hamilton Depression scale (HAMD-17). The final decision on patient inclusion was made by the psychiatrist, based on the following testing criteria: HAMD-17 ≥ 14, CES-D ≥ 27, and GAD-7 < 10. The HAMD-17 scale was considered the primary test, so a few patients whose CES-D or GAD-7 scale scores were slightly below threshold were still included in the study, based on doctor recommendations. No patients took medication prior to fecal sample collection.

Healthy volunteers (group HC) were chosen for the control group by the following criteria: no diagnosed psychiatric disease; CES-D < 18; GAD-7 < 5; no recent suicidal attempts through poisoning; no current eating disorders in the absence of competing diagnoses of post-traumatic stress disorder; no disorders connected to the use of psychoactive substances (alcohol, drugs); no history of severe traumatic brain injury; no infectious, autoimmune or somatic diseases that could affect molecular tests (for example, cancer, AIDS, or diabetes); and no anti-, pro- or prebiotic curation in the three months before fecal sample submission.

The final cohort included 38 healthy volunteers and 36 patients with depression. All information about both groups is presented in Table 1 and Appendix A.

### 2.2. Preparation and Sequencing of Metagenomic Samples

Fecal samples were taken using the standardized approach [39]. All samples were stored at a temperature of −80 °C. Total genomic DNA was extracted from fecal samples using the QIAamp PowerFecal Pro DNA Kit (Qiagen, Hilden, Germany) according to the manufacturer’s instructions. The concentrations of the extracted DNA were determined with Qubit dsDNA HS Assay Kit (Thermo Fisher Scientific, Waltham, MA, USA). The quality of DNA was checked on 0.8% agarose gel. For library construction, DNA was fragmented to an average size of about 350 base pairs (bp) using Covaris M220 (Covaris LLC., Woburn, MA, USA), and then paired-end libraries were produced using NEBNext Ultra DNA Library Prep Kit for Illumina (Illumina Inc., San Diego, CA, USA) using standard protocols. Quality control of the received DNA libraries was performed on an Agilent Bioanalyzer 2100 device (Agilent Technologies, Santa Clara, CA, USA) using a High Sensitivity Kit in accordance with the manufacturer’s protocol. Paired-end sequencing was performed on Illumina HiSeqX Ten (Illumina Inc., San Diego, CA, USA) using standard protocols. The sequenced samples in FASTQ format were deposited in the NCBI SRA database (BioProject ID: PRJNA762199).

### 2.3. Quality Control and Trimming

For quality control of the raw sequencing data, FastQC v0.11.5 was used [40]. After that, adapter sequences were clipped, and the quality of bases was improved with Trimmomatic v0.39 [41]. The low-quality bases (Q < 20) and sequences shorter than 50 base pairs were discarded from further analysis. Contamination with the human genome was removed by mapping the reads on the human genome (assembly version hg19) using bowtie2 v2.4.1 [42]. After all of the quality improvement processes, the final set of remaining reads still comprised more than 80% of the initial set. The average size of one sample was 6.34 gigabases and 38.9 million read pairs. Assembly of metagenomic reads into longer contigs was carried out with metaSPADes v3.14.1 [43]. Basic statistics for samples and their assemblies are shown in Appendix A.

### 2.4. Taxonomic Analysis

The taxonomic analysis of metagenomic reads was conducted using Kraken2 v2.1.2 with the setup ‘--confidence 0.5′ [44]. The relative abundance was calculated with Braken v2.6.2 [45]. The obtained taxonomic profiles were then analyzed on different levels: ‘Phylum’, ‘Genus’, and ‘Species’. Alpha diversity was assessed using Shannon’s index, calculated from the Kraken2 output using the Vegan package [46]. Beta-diversity was analyzed with the Vegan package using the Bray-Curtis distance matrix.

Differences in taxonomic profiles on ‘Phylum’, ‘Genus’, and ‘Species’ levels between the PwD and HC groups were identified using the Mann–Whitney U test with a significance threshold of *p* < 0.05 and an FDR (false-discovery rate) correction for multiple comparisons using the Benjamini–Hochberg method. Taxa identified in less than half of the samples were excluded from comparative analysis.

### 2.5. Development of the Reference Catalog

The general workflow for creation of the catalog is shown in Figure 1. Considering the important roles of GM metabolites in brain function and behavior, we assembled a list of those involved in depression pathology, based on previous studies [26]. Next, key enzymes involved in the production and degradation of these metabolites were selected, after analysis of published data. Then, we searched for the reference amino acid sequences for each selected enzyme according to two key criteria: (1) Have bacterial origin (preferably from human GM); (2) Be well described either in published papers, or in curated databases (KEGG [47], MetaCyc [48], UniProtKB/Swiss-Prot [49], etc.). Next, protein BLAST [50] and the NCBI protein database [51] were used to search for orthologs of the selected reference sequences in genomes of the 46 most common bacterial genera of the human GM. The full list of used bacterial genera is presented in Appendix A. New orthologs have been added to our previously developed catalog of gene orthologs for neuroactive compounds [52]. As a result, the final version of the assembled catalog included 1031 amino acid sequences of homologs for 101 enzymes (Appendix A).

### 2.6. Functional Metagenomic Analysis

For functional annotation, we analyzed the metagenomic assemblies using the updated algorithm introduced in our previous study (Figure 1) [52,53]. First, open reading frames (ORFs) were identified in the metagenomic contigs using MetaGeneMark [54]. The ORFs were then annotated using BLASTp [55] and the updated catalog, as a reference. The BLASTp alignments were sorted according to the following thresholds: identity ≥ 60% and relative alignment length ≥ 90%. After that, the abundances of each ORF were recovered by mapping the reads to the sequences of ORFs using bwa mem [56]. The readCounts were obtained with samtools idxstats [57] and then normalized using the Trimmed Mean of M-values (TMM) normalization method implemented through the edgeR package [58]. The relative abundances of the genes were compared between the two groups using the Mann–Whitney U test with a significance threshold of *p* < 0.05 and an FDR correction for multiple comparisons using the Benjamini–Hochberg method. Genes identified in less than half of the samples were excluded from analysis.

The contigs, which contain annotated ORFs, were analyzed using Kraken2 with the setup ‘--confidence 0.01’ in order to determine the bacterial origins of the ORFs. All sequences assigned as «Unclassified» were discarded from further analysis. Contigs without an assigned taxonomy on the ‘Species’ level were additionally aligned with BLASTn on bacterial genomes with a ‘complete’ assembly status from the RefSeq database [59]. Alignments with identity ≤ 90% were filtered out. Thus, metagenomic signatures, i.e., signature pairs (taxon; gene) with respective relative abundances, were obtained at the genus and species levels.

Pairs (taxon; gene) identified in less than half of the samples were excluded from comparative analysis. All remaining results were processed with the Mann–Whitney U test with a significance threshold of *p* < 0.05 and an FDR correction for multiple comparisons using the Benjamini–Hochberg method. Correlations between the relative abundance of each signature pair (at the ‘Species’ taxonomic level) and severity of the disease measured by the HAMD-7, CES-D, and GAD-7 scores were counted using Spearman’s correlation test using scipy.stats library (FDR correction with the Benjamini–Hochberg method) [60]. For statistically significant signature pairs of microbial features and the corresponding metadata, a multivariable analysis was performed using the MaAsLin2 package [61].

## 3. Results

### 3.1. Taxonomic Analysis of GM of Patients with Depression

Comparison of the alpha diversity of the GM of depressive patients and healthy volunteers did not show statistical significance, with *p*-values for both the ‘Genus’ and ‘Species’ taxonomic levels being higher than 0.8, according to the Wilcoxon test (Figure 2A,B). The beta diversity analysis, on the other hand, demonstrated a difference at the ‘Species’ level between the two groups (*p*-value = 0.001, PERMANOVA test, Figure 2D). The nMDS plots also showed that the samples in the PwD group were more dispersed than in the HC group. This, in combination with the Shannon metric, indicated that the GM of patients with depression diverged more than the microbiota of healthy volunteers.

After diversity analyses, the changes in abundances of identified taxonomic units were tested for statistical significance at three taxonomic levels: ‘Phylum’, ‘Genus’, and ‘Species’.

At the ‘Phylum’ level, Firmicutes showed a statistically significant decrease in the PwD group according to the Wilcoxon test, although the correction for multiple comparisons described this result as a false positive (corrected *p*-value = 0.41, Figure 2E, Appendix A).

At the ‘Genus’ taxonomic level, we managed to identify 16 taxa with statistically significant changes in relative abundances (FDR-corrected *p*-value < 0.05, Figure 2F, Appendix A). Among them were genera *Escherichia*, *Faecalibacterium*, *Lachnospira*, *Roseburia*, and *Rutenibacterium*, whose average abundance was higher than 0.5% of the average of all samples in both groups. Genera *Faecalibacterium*, *Lachnospira*, and *Roseburia*, which widely occurs in human intestinal microbiota, were significantly decreased in the PwD samples compared with the HC group (*p*-values of 0.004, 0.024, and 0.004, respectively), whereas genera *Escherichia* and *Rutenibacterium* were increased (*p*-values of 0.041 and 0.004, respectively). Despite showing statistically significant changes, genera *Faecalibaculum*, *Intestinibaculum*, *Kocuria*, *Lancefieldella*, *Pseudobutyrivibrio*, and *Trueperella* had very low relative abundances across the samples in both HC and PwD groups (approximately 10^−5^–10^−4^%).

We also identified 26 species with significantly different changes in abundances (FDR-corrected *p*-value < 0.05, Figure 2G, Appendix A). Six species, which had an average abundance higher than 0.5% and showed significant changes, corresponded to the five genera described above (Figure 2F). These included *Escherichia coli* (increased, *p*-value = 0.039), *Faecalibacterium prausnitzii* (decreased, *p*-value = 0.008), *Lachnospira eligens* (decreased, *p*-value = 0.035), *Roseburia hominis* (decreased, *p*-value = 0.035), *Roseburia intestinalis* (decreased, *p*-value = 0.003), and *Ruthenibacterium lactatiformans* (increased, *p*-value = 0.004). Also, four species of genus *Veillonella* showed a significant decrease: *Veillonella atypica* (*p*-value = 0.004), *Veillonella dispar* (*p*-value = 0.008), *Veillonella parvula* (*p*-value = 0.037), and *Veillonella* sp. T1-7 (*p*-value = 0.014, Appendix A).

### 3.2. Genes

After taxonomic analysis, metagenomic reads for each sample were assembled into contigs. Then, ORFs were predicted and annotated using the updated catalog of homologs. Relative abundances for the putative genes encoding enzymes were counted and compared, as described in the ‘Materials and Methods’ section. The genes encoding enzymes identified in more than half of the analyzed samples are presented in Figure 3, and complete results are listed in Appendix A. We found a statistically significant decrease in abundances in the PwD group in comparison with the HC group (FDR-corrected *p*-value < 0.05) for genes involved in the synthesis of arginine (argininosuccinate lyase), asparagine acid (asparagine synthetase asnA), GLU (glutamate synthase subunits gltB and gltD), spermidine (spermidine synthase), and in genes involved in the degradation of 17-beta-estradiol (estradiol 17-beta-dehydrogenase), degradation of serotonin for melatonin production (serotonin N-acetyltransferase), and linoleic acid conjugation (linoleic acid isomerase). Increased abundances with an FDR-corrected *p*-value < 0.05 were shown for genes encoding catalase, dihydroxyacetone formation (dihydroxyacetone phosphatase), transportation of GABA (gamma-aminobutyrate antiporter), degradation of GLU (glutamate mutase subunits glmE and glmS and methylaspartate ammonia-lyase), and histamine (histidine ammonia-lyase).

### 3.3. Metagenomic Signatures

Taxonomic analysis of contigs containing the annotated ORFs allowed the reconstruction of metagenomic signatures on the ‘Genus’ and ‘Species’ taxonomic levels (Figure 4, Appendix A, respectively).

During the comparison of the metagenomic signatures of PwD and HC groups on the ‘Genus’ level, it was revealed that the genes that had shown an overall significant decrease in abundances in the previous stage (Figure 3) originated from genera *Faecalibacterium*, *Coprococcus*, and *Roseburia* (Figure 4A). The signature approach additionally revealed the decrease of butyryl-CoA dehydrogenase (butyric acid formation) in the *Faecalibacterium* genus, glutamine synthetase in *Faecalibacterium*, and phosphotransacetylase (acetic acid formation) in *Coprococcus*, *Faecalibacterium*, and *Roseburia*. Statistically significant increases in abundances was observed for genes encoding butyrate kinase, gamma-aminobutyrate antiporter, glutamate decarboxylase, and serine hydroxymethyltransferase in genus *Alistipes*; however, these signature pairs were identified in less than 60% of the samples, and mostly corresponded to samples of the PwD group.

Analysis of the signature pairs on the ‘Species’ level revealed that all described changes in the genus *Faecalibacterium* corresponded with changes in the species *F. prausntizii* (Figure 4B). Another gene from the same species encoding serine hydroxymethyltransferase also decreased; however, on the ‘Genus’ level, the significance of the signature pair (serine hydroxymethyltransferase; *Faecalibacterium*) did not pass the threshold (FDR-corrected *p*-value = 0.0566, Appendix A). The signature pair (asparagine synthetase asnA; *Coprococcus comes*) had a *p*-value below threshold but was identified in only half of the analyzed samples.

After recreating the metagenomic signatures, we checked whether the abundances of signature pairs correlated with severity of depression measured by CES-D and HAMD-17 indexes (Appendix A). The correlation tests were conducted for signatures on both the ‘Genus’ and ‘Species’ taxonomic levels. The Spearman test showed very weak correlations with various signature pairs, although the FDR-corrected *p*-values easily passed the threshold value of 0.01. The strongest correlations with each of the examined indexes (with the absolute correlation coefficient being higher than 0.4 and lower than 0.5) were observed for signature pairs (asparagine synthetase asnA; *F. prausnitzii*), (glutamine synthetase; *F. prausnitzii*), (estradiol 17-beta-dehydrogenase; *F. prausnitzii*), and (linoleic acid isomerase; *F. prausnitzii*). All of these correlations were negative, which means that the decreased abundance of the corresponding gene was associated with increased severity of disease. The multifactor analysis in MaAsLin2 also did not identify strong correlations with indexes. However, it evidence for the correlations of signature pairs (glutamine synthetase; *F. prausnitzii*) and (estradiol 17-beta-dehydrogenase; *F. prausnitzii*) with severity of depression.

As most of the significant genes were identified in *F. prausnitzii*, we analyzed how the genes were distributed amongst the various strains of this species. For this, we downloaded all 276 genomic assemblies of *F. prausnitzii* available on the GenBank database (June 2022) and conducted a search using BLASTp and our reference catalog. The complete list of analyzed genomes is presented in Appendix A. According to the results of this analysis, most of the genes of interest were quite broadly presented across the *F. prausnitzii* species, being found in 47–93% of the analyzed genomes (Appendix A).

## 4. Discussion

Research on the role of GM in depression is still an evolving area of study. This work presents data on the taxonomic composition and metabolic potential of GM in patients with depression compared healthy controls from the Moscow region. We studied whole metagenomes that were obtained using next-generation sequencing (NGS) technologies, which allowed us to identify unique microbial signatures of patients with depression relative to healthy controls. These are the first data obtained using this approach with a focus on the gut microbial community in Russian patients with depression.

The results obtained during the taxonomic analysis of GM in the PwD and HC groups were partially concordant with the previous studies. There were no significant differences in alpha-diversity. This is consistent with numerous studies that have also reported no differences in alpha-diversity between MDD and control groups [62,63,64]. The differences between groups during the beta-diversity analysis at the ‘Species’ taxonomic level were also similar to other studies [65,66].

On the other hand, we identified no significant differences at the ‘Phylum’ level. Only Proteobacteria were at relatively higher levels in the PwD metagenomes. However, changes in the relative abundances of *Firmicutes*, *Bacteroidetes*, and *Proteobacteria* phylotypes have been previously shown [67,68,69].

Changes in the abundances of bacterial genera and species in our study are also supported by the published data on taxonomic differences in MDD microbiota. Genera *Faecalibacterium* and *Roseburia* were present in significantly lower abundances in the PwD metagenomes. Decreased levels of *Faecalibacterium* and *Roseburia* may contribute to disease pathology, as depression is associated with a mild chronic inflammatory response, and *Faecalibacterium* is regarded to have a strong anti-inflammatory effect in GM [70]. Genus *Escherichia,* which belongs to the Enterobacteriaceae family, was also significantly increased in the PwD metagenomes. Overgrowth of *Escherichia* bacteria could lead to gut inflammation and increase permeability of the gut wall, which in turn favors bacterial translocation, promoting systemic inflammation [71]. Clinical depression has been shown to be accompanied with an increase in the pro-inflammatory cytokine, interleukin, such as IL-1b and IL-6 [72,73]. Elevated abundances of Enterobacteriaceae in the gastrointestinal tract has been shown to induce behavioral and psychological changes in animals and humans [74,75].

In agreement with previous reports [76], an abundance of the genus *Bifidobacterium* was non-significantly increased in PwD metagenomes. Bifidobacteria are known to exhibit probiotic properties and, recently, it has been proposed as a ‘psychobiotic,’ given its ability to produce neuromodulators and influence gut–brain relationships through interactions with other commensal bacteria [77,78,79,80,81]. The reason for these contradicting results remains unclear and requires further research. 

Previous studies on changes in the GM of patients with MDD have had different outcomes concerning the genus *Bacteroides*, with some studies indicating a decrease in their abundance [82], whereas others have found an increase in abundance [76,83]. In our study, the genus *Bacteroides* was more abundant in the GM of depressed patients, although without statistical significance. Previously, it was hypothesized that species of *Bacteroides* produced neurotoxic compounds. Bacteria of this genus synthesize lipopolysaccharides (LPS), which have been shown to be important stimuli for neuroinflammation and development of neurodegenerative disease [84,85].

Another statistically significant change in the GM of the PwD group includes the decrease in abundance of the genus *Lachnospira* and increase of the genus *Ruthenibacterium*. *Lachnospira* was among the most abundant genera in the HC group along with *Roseburia* and *Faecalibacterium*. Previous studies have described a negative correlation between *Lachnospira* and severity of depressive symptoms [86]. *Lachnospira* may promote behavioral changes through interactions with neurotransmitter systems, which is also one of the primary interactions in the gut–brain axis [87]. Bacteria of genus *Ruthenibacterium* are pathogens and were observed in COVID-19 patients, who showed decreased levels of immune cells and refractory hypoxemia [88]. 

At the species level, *F. prausnitzii* was more abundant in the control group, which is similar with previous findings on MDD microbiota. This species is known to produce butyric acid and other SCFAs [89]. In addition, a recent study on preclinical tests demonstrated that the intake of *F. prausnitzii* (ATC 27766) improved behavior associated with anxiety and depression, suggesting that this strain can be used as a psychobiotic [90]. In our research, statistically significant decreases in average abundance were observed for species *Lachnospira eligens*, *Roseburia hominis*, and *Roseburia intestinalis* in the microbiota of patients with depression, whereas the species *Ruthenibacterium lactatiformans* and *Escherichia coli* showed elevated abundances. Species *R. intestinalis*, like *F. prausnitzii*, are among the most prevalent butyrate-producing commensal bacteria [91]. Metabolites of *R. hominis*, propionate and butyrate, can stimulate synthesis of melatonin in the intestine by increasing 5-HT levels and promoting p-CREB-mediated transcription of *Aanat* gene [92]. 

Using the analytical workflow developed in-house for the targeted profiling of microbial pathways involved in the metabolism of compounds known as depression biomarkers, we detected changes in the neurometabolic potential of the GM of depressed patients. In the PwD metagenomes, we observed a statistically significant decrease in the abundances of genes encoding enzymes involved in the production of amino acids (arginine, asparagine, glutamine, and GLU), linoleic acid conjugation, melatonin production, synthesis of spermidine, and 17-beta-estradiol degradation, and an increase in abundances of genes encoding catalase, enzymes involved in the formation of gamma-aminobutyrate antiporter and GABA, dihydroxyacetone, and degradation of glutamate and histamine. 

According to previous clinical studies, levels of arginine, asparagine, GLU, and TRP in the blood were decreased in patients with depression [93,94].

GLU and its metabolite GABA are known as key excitatory or inhibitory neurotransmitters in the neurovascular system, and dysfunctions in their signaling pathways are closely related to depression [95,96]. *Bacteroides* ssp., *Parabacteroides*, and *Escherichia* species, which were increased in the PwD group, are GABA producers. However, the increase in abundance of the gene encoding glutamate decarboxylase (production of GABA) in the PwD metagenomes was statistically insignificant.

Lowered SCFA formation levels may lead to increased gut permeability, inducing a “leaky gut” [97]. This loss in integrity can cause the migration of bacteria and their products through the mucosal membrane [98]. Butyrate, as well as melatonin, may potentially serve as sleep-inducing signaling molecules to enhance sleep [99]. More than 90% of patients with depression have sleep disorders, and a small number of patients complain of drowsiness [100]. SCFAs have also been noted to cause an eight- to ten-fold increase in serotonin production, at least in in vitro colonic mucosal systems [101]. 

Spermidine is the most abundant among the polyamines in the human brain. Exogenously administered spermidine helps in the treatment of brain diseases [102]. Spermidine is also involved in cognitive function [103]. MDD patients demonstrate more frequent cognitive disturbances than the general population [104]. Altered polyamine levels and altered expression of polyamine genes were observed in the cortical brain areas of suicide completers [105]. Hippocampal levels of putrescine, spermidine, and spermine were shown to be significantly decreased in a rat model of depression [106]. 

Depression is accompanied by increased oxidative stress [107]; thus, metabolites associated with depression can serve as markers for developing depressive disorders. Our data showed increased levels in genes for antioxidants catalase and glutathione, and decreased abundance of the gene responsible for linoleic acid conjugation [108]. 

The differences in gene abundances that we found describe overall changes taking place across the whole metagenome; however, identifying the metabolic signature, which provides more detailed information about a metagenome, was the key point of our study. The signature approach allowed us to connect the differences in taxonomy with the described genes. Alterations in taxonomy and gene abundances appeared to be consistent with one another. At the ‘Genus’ level, the observed decrease in levels of genes involved in the formation of acetic and butyric acids was significant in SCFA-producing *Faecalibacterium* and *Roseburia* genera, which also showed significant decreases during the taxonomic analysis. The decreased abundance of the gene involved in the degradation of 17β-estradiol (E2) also corresponded with a decrease in the genus *Faecalibacterium*. E2 is the most potent naturally circulating estrogen, which affects cognition, anxiety, depression, and the GM [109]. Genus *Faecalibacterium* was also found to be associated with sex hormone levels [110]. Other changes correlating with the decreased abundances in *Faecalibacterium* included genes involved in the production of arginine, asparagine, conjugated linoleic acid, GLU, glutamine, melatonin, and spermidine.

At the ‘Species’ level, the signature pairs revealed that each gene from the genus *Faecalibacterium* that had significantly decreased abundances corresponded with the species *F. prausntizii*. These data represent the neurometabolic signature of the GM of depressive patients and can be used as biomarkers.

Although we did not find any strong correlations in our tests, there was a connection between severity of disease and decreased abundances of the genes involved in the metabolic pathways of 17-beta-estradiol, asparagine, GLU, and conjugated linoleic. The relatively low correlation coefficients might have resulted from the heterogeneity of patient conditions in the cohort.

A limitation of the current study was the relatively moderate size of the sample. Testing more participants with stricter inclusion criteria for patient selection would provide stronger statistical evidence and allow for deeper digging into the characteristics and relations of GM functionality in patients with various subtypes of depression (for example, different severity levels of the disease). Moreover, we mainly focused on the identification of certain genes (neuroactive compounds and biomarkers of depression) from the reference catalog. Using a broader reference may lead to the identification of additional metabolites correlated with manifestations of depressive disorder. It should also be noted that research based on metagenomic sequencing can only reveal associations between the compositional and functional changes in microbiota and the host’s condition. Metagenomic sequancing does not inform about causal relationships or the direction of interactions in the microbiota–gut–brain axis. 

Another limitation is connected with the strain-specificity of some functional properties, as outlined in our study regarding the described genes in different strains of *F. prausnitzii*. The current algorithm to identify metagenomic signatures only allows it at the ‘Species’ taxonomic level. Future studies should describe the genomic characteristics of species strains, and the differences between healthy controls and depressed patients. Thus, it is essential that currently existing methodologies are improved to identify bacterial diversity at the ‘Strains’ taxonomic level.

Further transcriptomic, proteomic, and metabolomic analysis of the GM and analysis of the GM of the MDD population in other countries will provide a broader view on the compositional and functional changes of microbiota. These results can be used as biomarkers to develop diagnostic tools for assessing the progression of depressive disorders.

## 5. Conclusions

In summary, this whole metagenome study indicated compositional changes in the GM of patients with MDD and subsequent changes in its neurometabolic potential. Our data support a number of previous studies that demonstrated a decrease in beneficial bacteria, such as *Faecalibacterium*, *Roseburia*, and *Lachnospira*, along with an increase in abundances of opportunistic pathogens, such as *Ruthenibacterium* and *Escherichia* in the PwD group compared with healthy controls. These changes characterized the dysbiosis of GM and could be contibuting factors to the development of depression. Therefore, stimulating growth of diminished beneficial bacteria and reducing numbers of undesirable bacteria may be a strategy for the correction of GM in depressive patients. Psychobiotics and pharmabiotics with well-described positive properties and mechanisms of action could be used for this purpose. Their application could maintain the balanced composition of the GM and may improve the symptoms of depressive disorders.

One of the key features of this study was the use of the metagenomic signatures (gene; genus) and (gene; species) to identify potential biomarkers of GM associated with MDD. Our analyses revealed statistically significant decreases in the levels of genes encoding enzymes involved in the production of amino acids in arginine, asparagine, glutamate, glutamine acetic and butyric SCFAs, melatonin, spermidine, and conjugated linoleic acid in the GM of patients with depression. All of these genes were found in signature pairs with *F. prausntizii* and correlated with decreases in this species in the GM of the PwD group.

These results may assist scientists in further exploration of MDD pathogenesis. The described biomarker bacteria and their metabolites can be used to determine differences in the GM of individuals suffering from depression. The identification of changes in the GM may provide valuable information for future choices of treatment.

## Figures and Tables

**Figure 1 biomedicines-10-02162-f001:**
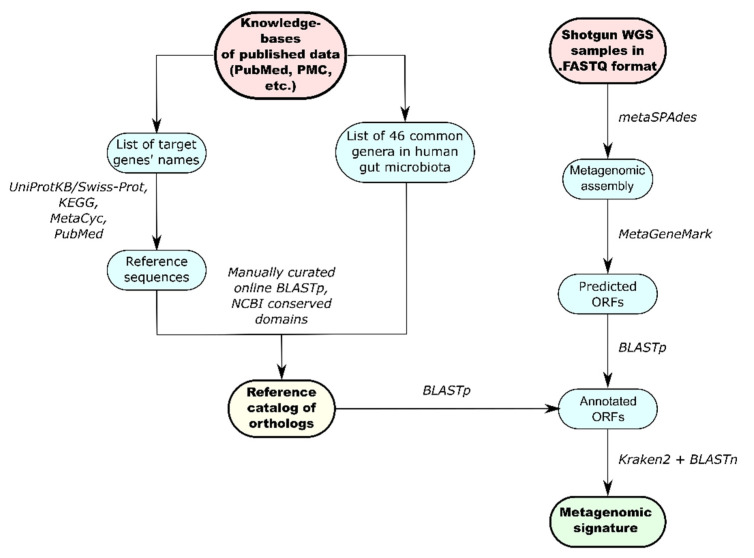
Schematic representation of the general bioinformatic workflow, including the algorithms for the reference catalog assembly and metagenomic signature reconstruction.

**Figure 2 biomedicines-10-02162-f002:**
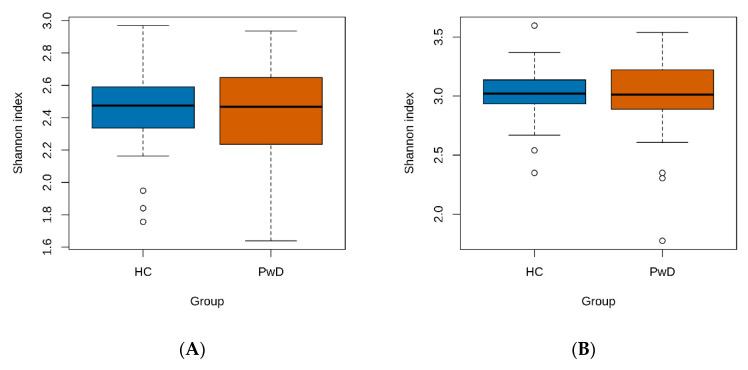
Diversity analysis of GM of MDD patients and healthy controls. (**A**,**B**): comparison of Shannon indexes between the HC and PwD groups at ‘Genus’ and ‘Species’ taxonomic levels, respectively. (**C**,**D**): nMDS plots describing beta-diversity between the HC and PwD groups at ‘Genus’ and ‘Species’ taxonomic levels, respectively. The points representing HC are colored in blue and PwD group are colored in red. Ellipses indicate confidence areas around the clusters of grouped samples. (**E**–**G**): taxa with median relative abundance > 0.5% detected in GM of the HC and PwD groups at ’Phylum’, ‘Genus’, and ‘Species’ levels, respectively. Median values of relative abundances are stated in percentages. Full taxonomic profiles are provided in Appendix A.

**Figure 3 biomedicines-10-02162-f003:**
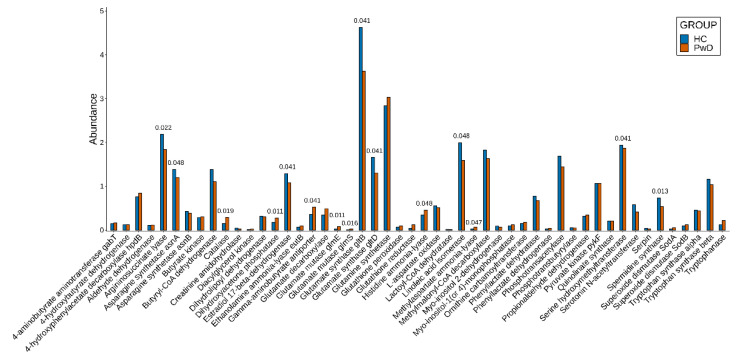
Changes in abundances of the genes in the PwD and HC groups. Only genes identified in more than half of the samples are shown on the graph. The values above the bars indicate the FDR-corrected *p*-values for genes with statistically significant differences in abundances.

**Figure 4 biomedicines-10-02162-f004:**
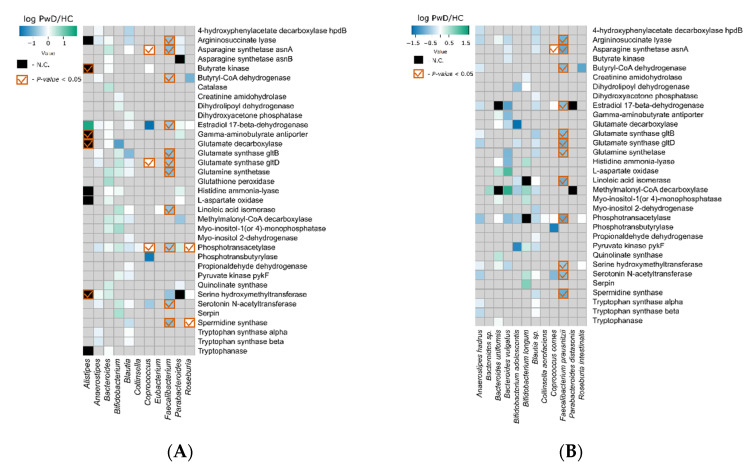
Metagenomic signatures describing changes in GM of patients with MDD at the (**A**) ‘Genus’ and (**B**) ‘Species’ taxonomic levels. Only signature pairs identified in the 46 most common bacterial genera (and corresponding species) of the human GM and found in more than 50% of the samples are presented. The color gradient represents the logarithm of ratios of median abundances in the PwD and HC groups. The green color gradient describes the increase in abundances of the signature pairs, and the blue color describes the decrease in abundances. The ‘N.C.’ values (colored in black) represent non-countable ratios, where the median abundance in the HC group was 0. The white color represents ratios where the median value in the PwD group was 0 (logarithm is undefined). Signature pairs with statistically significant changes in abundances are indicated with a red tick. More detailed information is provided in Appendix A.

**Table 1 biomedicines-10-02162-t001:** Characteristics of the final cohorts for study.

Sex, Male/Female	HC Group	PwD Group
	19/19	19/17
Age, y.o. (average, (min:max))	34, (18:54)	30, (18:53)
BMI, kg/m^2^ (average, (min:max))	24, (18:39)	22, (16:34)
CES-D (average, (min:max))	5, (0:17)	31, (21:52)
GAD-7 (average, (min:max))	1, (0:10)	7, (2:10)
HAMD-17 (average, (min:max))	1, (0:8)	21, (14:28)

## Data Availability

The metagenomic samples used in this research are available on the NCBI SRA database under BioProject ID PRJNA762199.

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
