# Peer review of "Alterations of the Composition and Neurometabolic Profile of Human Gut Microbiota in Major Depressive Disorder"

_biomedicines, 2022, doi:10.3390/biomedicines10092162_

Round 1
Reviewer 1 Report
Kovtun et al. identified the changes in gut microbiota and microbial metabolites and pathways that could contribute to depression. Whereas the study seems comprehensive, there are major concerns that should be addressed:
Introduction:
There is evidence of microbiota crosstalk with other cell types besides immune cells that are not mentioned in the introduction.
The last paragraph of the introduction is an overall of the study and justification, which can be substantially shortened and partially included in the materials and methods and the results sections.
Materials and methods:
It is not reported whether patients and/or controls were taking any medication.
It is not completely clear how data on amino acids, genes and enzymes is obtained. The authors could actually analyze the faecal samples in search of bacterial metabolites and levels of different amino acids.
Results:
The information in the result section should summarize the figures/tables instead of providing a detailed description of what can be obtained from the tables. Much of the results could be represented in figures instead of tables (and/or move the tables as supplementary material, i.e. tables 2, 3, 4, 5 and 6). Figure 2 and 3 could be combined. Some labels on the axes of figure 4 are missing and the font size is too small to be read. Ordering the results in this figure in descending order and shortening the list of genes would improve its visualisation.
In section 3.2, results regarding indexes are usually expected before a detailed description of the genus and species is found.
Discussion:
This section is too long and should summarize the results without getting into detailed descriptions. In its current version, the discussion section is not very well structured and could be easier to follow if topics of interest are identified and discussed at once and separately (e.g. gut permeability, inflammation, SCFAs, amino acid metabolism, serotonin and effects on neurons). Information should be linked more adequately.
Conclusions:
There is no clear point or conclusion stated here and the section is way too long. There is no clear "take-home" message with clinical significance.
Other concerns:
Symbols are missing (line 86) and scientific annotations need to be fixed (line 90).
Section 3.1. and the first paragraph of 3.2 of the results section belong to materials and methods section. Also, to make this information more visible, it could be depicted in figure 1.
Authors should consider a different colour combination in some of the figures that could be more appropriate for daltonic audience.
The authors should exclude subjective impressions from the text and try to report and discuss their results in an appropriate way (e.g. suppress expressions such as " These data are difficult to explain."). Formal English should preferably used along all the text.
Author Response
Dear reviewer,
Thank you for your suggestions and comments on our manuscript. Please, find below the point-by-point replies and comments on what has been changed. We really hope that these changes have improved our manuscript.
Sincerely,
Alexey Kovtun, corresponding author.
Introduction:
- There is evidence of microbiota crosstalk with other cell types besides immune cells that are not mentioned in the introduction.
We have described the main mechanism of communication of gut microbiota with its host in the specific paragraph (lines 68-83). We have included the functional aspects gut microbiota, which can possibly play essential role in the development of depression, such as immuno- and neuromodulation.
- The last paragraph of the introduction is an overall of the study and justification, which can be substantially shortened and partially included in the materials and methods and the results sections.
We have shortened the last paragraph, leaving only the key point of our study.
Materials and methods:
- It is not reported whether patients and/or controls were taking any medication.
No patients had taken any medication before the fecal samples’ collection. We have added the corresponding sentence (line 140 of section 2.1).
- It is not completely clear how data on amino acids, genes and enzymes is obtained. The authors could actually analyze the faecal samples in search of bacterial metabolites and levels of different amino acids.
As this question can apply to either catalog assembly, or the data analysis workflows, I’ll answer both. The data on which amino acids and genes were essential for our consideration and further inclusion into the reference catalog, a deep analysis of the published data had been conducted (research articles, reviews, databases, like KEGG, MetaCyc, Uniprot, etc.). Regarding the analysis workflow, data on possible presence of the genes encoding enzymes of our interest have been obtained by the straightforward analysis of the NGS data (that is why we used the whole metagenome data). First, we’ve assembled the metagenomic reads into longer contigs. Next, we’ve predicted putative genes (open reading frames) with the specialized tool (MetaGeneMark). Finally, we’ve annotated them using the developed catalog. We have also improved the scheme describing our workflow and included the algorithm of the catalog’s assembly (Figure 1).
Results:
- The information in the result section should summarize the figures/tables instead of providing a detailed description of what can be obtained from the tables. Much of the results could be represented in figures instead of tables (and/or move the tables as supplementary material, i.e. tables 2, 3, 4, 5 and 6). Figure 2 and 3 could be combined. Some labels on the axes of figure 4 are missing and the font size is too small to be read. Ordering the results in this figure in descending order and shortening the list of genes would improve its visualisation.
We have completely reworked the data representation. All large tables have been replaced with bar charts and heatmaps. The description of the results has been tuned correspondingly.
Graphs describing the results of the taxonomic analysis have been united into a single cluster of five figures. Few changes have been done to the description of the taxonomic analyses, as the bar charts don’t include information on the statistics and representation of each taxon among the cohort. We find it essential and, thus, left it in the text.
The description of the genes has been reworked in order not to be too descriptive and to focus more on the potential functional changes. We have also did our best to make the fonts larger (they are now 16 pt instead of 14 pt) on the corresponding figure and placed the legend inside the plot, but we’ve still decided to leave the list of the genes almost untouched (only two genes identified in exactly 50% of samples and showing no significance were excluded). The reason for this decision lies in the metagenomic signatures. They include many of these genes, and some showed significant change at taxonomic resolution, while not passing the thresholds during the analysis of the overall amounts of genes. For example, genes, encoding butyryl-CoA dehydrogenase and phosphotransacetylase. Still, the original figure, which can be accessed from the journal’s website, is of a large size and great resolution.
The section about the metagenomic signatures has also been reworked along with their representation. Now there are two figures describing changes at ‘Genus’ and ‘Species’ taxonomic levels, respectively, instead of one summarizing figure and two tables.
- In section 3.2, results regarding indexes are usually expected before a detailed description of the genus and species is found.
We have it placed before the comparative analysis now.
Discussion:
- This section is too long and should summarize the results without getting into detailed descriptions. In its current version, the discussion section is not very well structured and could be easier to follow if topics of interest are identified and discussed at once and separately (e.g. gut permeability, inflammation, SCFAs, amino acid metabolism, serotonin and effects on neurons). Information should be linked more adequately.
We have rewritten this section in order to make it more readable and understandable.
Conclusions:
- There is no clear point or conclusion stated here and the section is way too long. There is no clear "take-home" message with clinical significance.
We have rewritten this section in order to make it short and straight-to-the-point clear.
Other concerns:
- Symbols are missing (line 86) and scientific annotations need to be fixed (line 90).
We have fixed it.
- Section 3.1. and the first paragraph of 3.2 of the results section belong to materials and methods section. Also, to make this information more visible, it could be depicted in figure 1.
We have replaced this section closer to the actual functional analysis and included the illustration to the Figure 1.
- Authors should consider a different colour combination in some of the figures that could be more appropriate for daltonic audience.
We have fixed it using the built-in R brewer palettes recommended for colour-blind people (library RColorBrewer). For very complex pictures like 2F and 2G, which had more than 8 positions to be coloured, we have used a combination of three palettes, namely ‘Paired’, ‘Dark2’ and 'Set2'. They are quite contrasting and don’t have intersecting colours.
- The authors should exclude subjective impressions from the text and try to report and discuss their results in an appropriate way (e.g. suppress expressions such as " These data are difficult to explain."). Formal English should preferably used along all the text.
We did our best to fix this.
Reviewer 2 Report
The article is well-written, easy to understand and well-structured. The authors approached a high interest topic and the study is extremely interesting.
Nevertheless, I would have the following suggestions:
- the authors should state more clearly the limitations in a distinct paragraph of this study since I consider that there is more than one limitation
- they should also mention the strenghts of the study in the discussions section
- the authors shoud not mention 'in summary' in the discussions section since this stands for a conclusions and if they consider the statement to be a conclusion, it should be moved in the conclusions section
- the conclusions section is a bit to detailed, the authors should focus on mentioning only the conclusions of their study and perhaps to state the gaps that deserve future studies
Author Response
Dear reviewer,
Thank you for your suggestions and comments on our manuscript. Please, find below the point-by-point replies and comments on what has been changed. We really hope that these changes have improved our manuscript.
Sincerely,
Alexey Kovtun, corresponding author.
- the authors should state more clearly the limitations in a distinct paragraph of this study since I consider that there is more than one limitation
We have described it in the ‘Conclusions’ section.
- they should also mention the strengths of the study in the discussions section
The key strengths of our study are mentioned in ‘Introduction’ and ‘Discussion’ sections, which, in our opinion, is the utilization of the whole metagenome sequencing and the metagenomic signature approaches.
- the authors should not mention 'in summary' in the discussions section since this stands for a conclusions and if they consider the statement to be a conclusion, it should be moved in the conclusions section
We have fixed this issue.
- the conclusions section is a bit to detailed, the authors should focus on mentioning only the conclusions of their study and perhaps to state the gaps that deserve future studies
We now have our ‘Conclusion’ completely reworked. We hope that it has become clearer and more straight-to-the-point.
Round 2
Reviewer 1 Report
The authors have substantially improved the quality of the manuscript by re-structuring the text and adding new figures. However, some major concerns remain:
* The authors did not answer why bacterial metabolites and levels of different amino acids were not measured on faecal samples to validate their findings.
* Since the submitted new version of the manuscript kept track of the changes made, it is not fully clear which figures are included in its final version. Still, in figure 2, why do relative abundance bar plots not reach 100% if they are intended to show relative abundance? Also, in figure 4, why black and white boxes representing non-countable ratios and ratios where the logarithm is undefined, respectively, can have a white or red tick?
Some other figures need further clarification (e.g. what do the ovals on nMDS exactly mean?) or re-coding for colours (e.g. Shannon index and nMDS plots in figure 2, also figure 3).
* Sentences between lines 111 and 121 seem vague, while references and clear examples are lacking.
* A summary figure depicting all the findings of the study would add value to the manuscript.
Minor concerns:
* There are many acronyms that are yet to be clearly identified. Please also find a better acronym for depression as PD could be mistaken for Parkinson's disease.
* English should still be more formal (e.g. avoid contractions such as "didn't" and use "did not" instead).
Author Response
Dear reviewer,
Thank you for your suggestions and comments on our manuscript. Please, find below the point-by-point replies and comments on what has been changed. We really hope that these changes have improved our manuscript.
Sincerely,
Alexey Kovtun, corresponding author.
* The authors did not answer why bacterial metabolites and levels of different amino acids were not measured on faecal samples to validate their findings.
The study of changes in the number of biomarker bacterial metabolites in the feces of patients with depression will be carried out as part of the following project in the near future. We have included the corresponding clarification in the “Conclusion” section.
* Since the submitted new version of the manuscript kept track of the changes made, it is not fully clear which figures are included in its final version. Still, in figure 2, why do relative abundance bar plots not reach 100% if they are intended to show relative abundance? Also, in figure 4, why black and white boxes representing non-countable ratios and ratios where the logarithm is undefined, respectively, can have a white or red tick?
Some other figures need further clarification (e.g. what do the ovals on nMDS exactly mean?) or re-coding for colours (e.g. Shannon index and nMDS plots in figure 2, also figure 3).
Unfortunately, we didn’t quite understand the concern with the inclusion of the figures. All of them are a replacement for the corresponding tables and are included in the main text of the final version of the manuscript.
Figures 2E-G show only abundant taxa with median relative abundance >0.5%. It has already been mentioned in the figure legend. The full taxonomic profiles can be found in the corresponding supplementary figures. We have slightly rewritten the legend, so this point goes first.
In Figure 4 this graphical decision has been made just to gain a more contrasting picture (white ticks are easier to find on the black background, then the ‘red’ ones). Since this decision appeared to be confusing, we have re-painted all of the ticks into red.
The ovals for the nMDS figures indicate confidence areas around the clusters of grouped samples. We have added the corresponding clarification to the legend.
The colors for the mentioned figures have been recoded. We hope that they have become more colorblind-friendly.
* Sentences between lines 111 and 121 seem vague, while references and clear examples are lacking.
We have added a reference.
* A summary figure depicting all the findings of the study would add value to the manuscript.
Unfortunately, we didn’t have enough time to figure out and create such picture. Nevertheless, the metagenomic signatures can be pretty much considered as the summarizing picture, as they combine the information about both taxonomy and the genes.
Minor concerns:
* There are many acronyms that are yet to be clearly identified. Please also find a better acronym for depression as PD could be mistaken for Parkinson's disease.
We understand how misleading acronyms sometimes can be and made sure that they are clarified in the text and can be easily recovered. The acronym for the patients with depression is also mentioned in the “Materials and methods” section as a special-for-current-article one, and it is normally written as “PD group”, or “PD samples” along the whole text.
* English should still be more formal (e.g. avoid contractions such as "didn't" and use "did not" instead).
We have fixed this issue.
Reviewer 2 Report
Thank you for following the suggestions. Nevertheless, the limitations should be mentioned at the end of discussions section and not in the section of conclusions.
The conclusions section is meant for the description of your results in fact the conclusion section should summarize the main findings of your research.
Author Response
Dear reviewer,
Thank you for the kind response to the made updates.
Nevertheless, the limitations should be mentioned at the end of discussions section and not in the section of conclusions.
We have replaced the limitations of the study to the end of the “Discussion” section and rewritten the “Conclusion”.
Sincerely,
Alexey Kovtun, corresponding author.
Round 3
Reviewer 1 Report
The authors have properly addressed all the comments so I have no further concerns.
Author Response
Thanks for your comment.